# Segment, Compare, and Learn: Creating Movement Libraries of Complex Task for Learning from Demonstration

**DOI:** 10.3390/biomimetics10010064

**Published:** 2025-01-17

**Authors:** Adrian Prados, Gonzalo Espinoza, Luis Moreno, Ramon Barber

**Affiliations:** RoboticsLab, Universidad Carlos III de Madrid, 28911 Madrid, Spain; gespinoz@pa.uc3m.es (G.E.); moreno@ing.uc3m.es (L.M.); rbarber@ing.uc3m.es (R.B.)

**Keywords:** learning from demonstration, imitation learning, movement primitives, Gaussian mixture models, Gaussian process

## Abstract

Motion primitives are a highly useful and widely employed tool in the field of Learning from Demonstration (LfD). However, obtaining a large number of motion primitives can be a tedious process, as they typically need to be generated individually for each task to be learned. To address this challenge, this work presents an algorithm for acquiring robotic skills through automatic and unsupervised segmentation. The algorithm divides tasks into simpler subtasks and generates motion primitive libraries that group common subtasks for use in subsequent learning processes. Our algorithm is based on an initial segmentation step using a heuristic method, followed by probabilistic clustering with Gaussian Mixture Models. Once the segments are obtained, they are grouped using Gaussian Optimal Transport on the Gaussian Processes (GPs) of each segment group, comparing their similarities through the energy cost of transforming one GP into another. This process requires no prior knowledge, it is entirely autonomous, and supports multimodal information. The algorithm enables generating trajectories suitable for robotic tasks, establishing simple primitives that encapsulate the structure of the movements to be performed. Its effectiveness has been validated in manipulation tasks with a real robot, as well as through comparisons with state-of-the-art algorithms.

## 1. Introduction

Currently, robotics must adapt to a wide variety of environments and situations. One approach to enable robots to understand how to perform required tasks in new environments is through the learning process. Since these environments are designed to accommodate human structures, one effective learning strategy is the use of Learning from Demonstrations (LfDs) techniques [1,2,3]. This approach relies on information provided by a teacher who demonstrates the characteristics necessary to comprehend and reproduce the movements required to perform a task. Leveraging this information, LfD algorithms can learn to replicate and adapt the underlying motion information to novel environments, thereby enhancing task generalization [4].

Within this field, there are various methods to address this problem. Some are derived from Reinforcement Learning (RL) processes, aiming to infer state–action relationships without explicitly defining a cost function, commonly referred to as Inverse Reinforcement Learning (IRL) [5,6]. Other approaches utilize Deep Learning techniques [7]. The methods employ discretizers such as Neural Networks to acquire the underlying skills of a task [8] or Transformers-based techniques to address high-dimensionality problems that may create distractions in the learning model [9]. However, among the most widely adopted methods are those that encode tasks through complex movements, which are both easily replicable and interpretable by algorithms. These include techniques based on movement primitives [10,11].

The use of movement primitives allows tasks to be understood as simple movements that can be replicated and understood by learning algorithms. If the tasks are simple, they can be represented by basic actions or primitives that involve movements easily reproducible by robotic arms. However, for more complex tasks, such as setting a table [12], a sequence of movement primitives may be required, connecting one to another to achieve these more complex tasks [13]. This approach of combining different simple movement primitives offers multiple advantages for task resolution. For instance, certain movement primitives may be common to different types of actions. This allows a single set of primitives to be used in various scenarios, enabling the reuse of acquired knowledge.

The acquisition of such movement primitives presents a clear limitation: the ability to obtain this information in an automatic and generalized manner for different tasks without requiring a manual selection process by the user. One way to address these challenges is through the creation of synthetic data that generalize tasks automatically [14], generating data that are similar in terms of energy efficiency for a specific movement primitive. However, these methods may face challenges when applied to highly complex tasks and are primarily used to acquire simple primitives. Therefore, in this work, we present an approach aimed at solving this problem in a more general manner by providing an algorithm for the automatic segmentation of unlabeled demonstrations, presented in Figure 1. This algorithm can account for different types of information in its segmentation process through a probabilistic approach, as well as autonomously create libraries of simple primitives based on these segmentations.

In this work, we focus on addressing two primary aspects. The first is the segmentation of complex and lengthy tasks into smaller segments that can be easily learned and encapsulated within libraries of similar movements. The second aspect involves the generation of movement libraries based on the similarity between movements, enabling the creation of reusable libraries. These libraries eliminate the need to re-teach similar movements, thereby facilitating their combination to generate new, complex tasks.

To achieve these objectives, we first address the segmentation process. We present an algorithm capable of automatically segmenting multiple demonstrations by considering various types of information obtained during the demonstrations, such as velocity and wrist force. Each type of information is temporally treated as a probability distribution of segmentation points. These probabilities are determined based on abrupt changes in data continuity, evaluated using the third derivative. Subsequently, a probabilistic combination process is applied to mitigate issues of over-segmentation that can arise from heuristic-based methods. This approach allows for the merging of multiple very short segments by grouping them according to similar probabilities and defining them as a single segment.

After obtaining the segmentation of the demonstrations, the algorithm proceeds to generate the corresponding movement libraries. To achieve this, the proposed algorithm evaluates the energy cost (based on the probabilistic similarity between the segments) required to transform one segment relative to the others. This enables the construction of a transportation cost map, allowing the grouping of different segments with those that exhibit the lowest energy cost, thereby clustering similar segments. Additionally, the method incorporates a probabilistic representation of the demonstrations as a whole. By accounting for variations in segment uncertainty, this approach facilitates the evaluation of movements that are similar but not identical, which is common when data are collected from multiple users.

A critical premise of our method for generating movement primitives is their complexity. The algorithm considers the tendency of heuristic methods to over-segment trajectories in a suboptimal manner, often resulting in the creation of primitives that lack meaningfulness and introduce irrelevant information. This, in turn, complicates the generation of complex tasks. To address this, we apply a probabilistic inference process that iteratively identifies groups of segments with similar weights. This ensures an appropriate correlation between the complexity of the segmented information and the complexity of the resulting libraries. Through this inference process, we generate a series of movement libraries that are sufficiently complex to represent specific movements, while remaining simple enough to be learned by LfD algorithms without incurring excessive computational costs. This approach effectively groups movement libraries, clustering common movements that are highly similar into a single library, while assigning more specific or complex movements to distinct libraries. The energy cost metric plays a crucial role in this process, as it quantitatively evaluates the cost or complexity between different segmentations. This enables the identification of movements that are most similar to each other, thereby facilitating efficient grouping and library creation.

In summary, this work introduces the **Segmentation and Grouping Model (SeGM)** algorithm, which enables the automatic segmentation of human demonstrations and the creation of movement libraries based on the probabilistic cost between these segments. To validate the algorithm, we performed tests in both 2D and 3D simulations using tasks directly captured from a mobile manipulator robot. We evaluated the algorithm with a letter segmentation task to learn primitives using SeGM and subsequently reuse that knowledge to write other words with the learned primitives. A similar test was performed for a 3D task involving the pick-and-place of various objects, where the task generation order was modified using the segmented and learned primitives.

The paper is organized as follows. Section 2 provides a review of state-of-the-art algorithms for the problem of automatic segmentation applied within the context of Learning from Demonstration. Section 3 offers a detailed description of the proposed algorithm, focusing on each of its components. Section 4 presents an example of the complete operation of the algorithm in both 2D and 3D, comparisons with different segmentation algorithms, and experiments applying the motion libraries with a real robot. Finally, Section 5 outlines the conclusions drawn from this work, as well as potential future research directions. This work has a code associated in https://github.com/AdrianPrados/Segmentation-and-Grouping-Model, accessed on 18 December 2024.

## 2. Related Work

The understanding of complete human tasks has been one of the most extensively studied research focuses within Imitation Learning algorithms. Such tasks are commonly referred to as Long-Horizon tasks [15,16]. The underlying principle of these methods is that, given the complexity and duration of human tasks, learning algorithms should be capable of acquiring the task through small segments of movements. This approach enables tasks to be decomposed into simpler subtasks that can be easily understood by learning algorithms while also being highly generalizable.

Many of these techniques focus on decomposing movement information through various LfD methods. One widely adopted technique is the application of Gaussian Mixture Models (GMMs) [17], which are typically employed to probabilistically model a path as a sequence of Gaussian distributions, thereby enabling the replication of the learned task. Within this field, several algorithms have been developed, such as the one presented in [18], which introduces a task sequencing algorithm encoded through hidden semi-Markov models. Another example utilizing GMMs is presented in [19], where an LfD algorithm is proposed that incorporates an “action layer” to enable data segmentation using a Gaussian mixture approach. This segmentation process facilitates the grouping of distinct movements into similar primitive tasks that follow a predefined structure determined by these Gaussian models. In both methods, the algorithms cluster demonstrations that may exhibit slight variations but represent relatively simple data corresponding to a single type of task. However, these algorithms are not capable of independently segmenting complex tasks; they can only group them into primitives. Furthermore, these methods have not been tested with movements exhibiting significant differences in structural form, as the movements analyzed are typically relatively similar.

Other approaches, based on Task-Parameterized Gaussian Mixture Models (TPGMMs) [14,20], are used for learning complex tasks. For instance, in [12], an algorithm is proposed to generate complex trajectories by decomposing a complex task, such as setting a table, into simpler subtasks. This work applies divergence-based techniques [13] to evaluate and identify the most relevant parameters for autonomously solving the task. A major limitation of these methods is their reliance on a predefined library of previously decomposed movements and their down-to-top trajectory generation process (i.e., creating complex movements from known simpler ones). In contrast, the objective of our approach is to enable the algorithm to automatically decompose complex tasks into simpler ones (top-to-down) and subsequently rearrange these into new complex tasks. Within the TPGMM framework, another example is presented in [15], where an algorithm observes a scene and decomposes the demonstrated tasks into a skill library that can later be reused for new tasks. This algorithm requires five demonstrations and relies on identifying decomposition points through relevant context provided by objects observed via a camera. In the presented work, however, there is no need for a physical context of demonstrations. The algorithm autonomously and probabilistically identifies relevant points and can operate with *N* demonstrations.

Within the domain of learning with movement primitives and their derivatives, such as Dynamic Movement Primitives (DMPs) [21] or Probabilistic Movement Primitives (ProMPs) [22], the use of segmented information in small subtasks is essential for proper functioning. Consequently, this area has seen extensive algorithmic development. An example of such processes is presented in [23], where an algorithm based on Probability-based Movement Primitives (PbMPs) is proposed. This method enables the identification of changes in curvature, allowing the approximation of each subtask composing the complete task to the users’ demonstrations. While this approach facilitates the subdivision of simple tasks, its effectiveness in highly complex tasks remains unproven. Additionally, it relies on a fully heuristic method, often resulting in over-segmentation into unnecessary subtasks. Other methods address the multi-step learning of tasks through a combination of Task and Motion Planning (TASM) with an optimal control formulation of DMPs [24]. This approach provides solutions that establish the logical order of subtasks required to complete a complex goal. In this case, the TASM algorithm generates each subtask without relying on a predefined library. However, it lacks generalization capability for similar cases, creating specific movement primitives that only adapt marginally to local model variations. Other works leveraging DMPs focus exclusively on the clustering aspect, assuming a predefined library of previously generated primitives. For example, in [25], an Expectation–Maximization (EM) process is employed to estimate the duration and final position of a partial trajectory, allowing automatic selection of one of the previously learned primitives.

In recent years, a growing area of research has focused on achieving unsupervised task segmentation, enabling its application in various learning algorithms that rely on movement libraries. One of the most significant contributions in this domain is presented in [26], which introduces an algorithm that simultaneously performs task segmentation into movement primitives while learning the required primitives for encoding. This approach enhances both the quality of segmentation and the skill learning process. To achieve this, the algorithm employs specific insights derived from a heuristic, evaluated through probabilistic techniques. An improvement to this algorithm is proposed in [27], where the dependence on the heuristic process is reduced, thereby refining the segmentation and learning framework. Additionally, the work in [28] presents a framework for generating movement primitives for Learning from Demonstration (LfD). This framework performs multimodal and automatic segmentation while organizing clusters of similar skills. The process leverages a probabilistic approach to manage the multimodal elements and employs clustering techniques based on elastic maps [29], facilitating dimensionality reduction. Another noteworthy method is described in [30], which combines segmentation and gesture recognition by analyzing the spatiotemporal persistence of the data and estimating a ranking of weights. One of the most recent works in this area is presented in [31]. This work introduces a method for segmentation and gesture recognition applied to minimally invasive surgical techniques. The algorithm is based on an analysis method using the Levenshtein distance to evaluate the curves generated by the segmentation process.

Unlike all the works presented in this section, our work introduces an algorithm that performs both the segmentation and clustering processes for the creation of movement libraries for LfD algorithms. This algorithm includes an initial automated segmentation phase, where critical points are identified using a heuristic that segments probabilistic data. This process is followed by a merging technique based on probabilistic weights to mitigate the over-segmentation issue. Secondly, the algorithm incorporates an automated clustering process by comparing probabilistic cost estimations. This estimation determines the cost required to make two segments exactly identical. The main contributions of our work are presented below.

Automated segmentation process capable of working with *N* demonstrations. It does not require any initial parameters or input for its operation. This segmentation utilizes a combined heuristic that leverages abrupt changes in acceleration to detect peaks in trajectories and smoothness variations through the application of jerk analysis. A probabilistic combination of segments, based on GMMs, is employed to prevent over-segmentation.Multimodal system integration. The algorithm can capture not only position information but also other relevant factors for manipulation tasks, adapting the relevant points for automated segmentation accordingly.Automated clustering process without the indication of clusters needed of initial parameter. This step uses an energy cost approach through Gaussian Processes (GPs) [32], which encapsulate the information of each analyzed segment and store those segments that are energetically similar.Comparative analysis. Experiments were conducted against other automated segmentation algorithms, demonstrating the approach’s utility through tests in both 2D and 3D environments using robotic platforms. Those experiments are presented in https://www.youtube.com/watch?v=A3m2sbUI5F0, accessed on 17 December 2024, and the code is in https://github.com/AdrianPrados/Segmentation-and-Grouping-Model, accessed on 18 December 2024.

## 3. Method Description

### 3.1. General Description

The algorithm presented in this work enables the subdivision of a complex task, allowing the identification of the basic or primary movements required for generating a complex trajectory. It also facilitates clustering for values with similar structures, which can thus be understood as the same basic movement. The algorithm focuses first on addressing the problem of subdividing into basic motion primitives that are sufficiently distinct to be differentiated from one another. To achieve this, we assume that each observed demonstration d∈D can be represented as a segmentation of that demonstration s∈d. This allows the entire set of demonstrations *D* to be expressed as the concatenation of different segments *s* from each demonstration within the initial dataset.

To obtain the segments into which our demonstrations are divided, it is necessary to determine, what we have called, *critical cutting points P*. These points specifically indicate where each segment begins and ends within the complete demonstration. To achieve this, a specific heuristic has been developed, detailed in Section 3.2. This heuristic identifies relevant points within the demonstrations based on the structure of the demonstration data. It is important to note that by using a heuristic, we assume it tends to generate over-segmentations in the data. Since the goal is to achieve a segmentation sufficient to extract the primitive movements of a complex task while minimizing the number of segments to avoid redundancies, additional measures must be taken to address over-segmentation. To this end, the SeGM employs a segment fusion algorithm based on probabilistic similarity using GMMs for that purpose. This approach calculates the probabilistic weight of each segment and groups those with similar values. This effectively eliminates over-segmentation implicitly. Section 3.2 provides a detailed explanation of this process, focusing on the mathematical definitions involved and presenting examples of the procedure.

Once the segments have been correctly identified, the algorithm performs a similarity-based assignment of segments. This enables the grouping of similar trajectories into movement primitives, thereby reducing the redundancy of libraries that may be considered identical. For this process, the algorithm employs what we have called *energy cost* estimation, required to transform one segment into another, considering both their shape and size. Gaussian Processes (GPs) are utilized to probabilistically estimate the variability of a single data point or a set of segments from different demonstrations. This approach establishes a cost map necessary to adjust the structure of each GP across different groups of segments. The mathematical definitions underlying these processes are detailed in Section 3.3. Once the costs have been estimated, the algorithm groups the different segments comprising the tasks into movement primitives (or skills). These primitives can later be reorganized, allowing for task generalization without the need to learn new tasks from human demonstrations. Additionally, the clustering algorithm enables a comparison to determine whether previously known primitives exist or if it is necessary to learn a new skill. Section 3.3 provides a detailed explanation of the process used to generate the movement library through probabilistic costs. Additionally, Figure 2 offers a comprehensive visualization of the processes within each part of the algorithm, presenting a visual schematic of all the mathematical operations involved in the segmentation process and the creation of the movement primitive library.

### 3.2. Segmentation of the Data

Starting from a series of trajectories $D=[d_{1},d_{2},\ldots ,d_{N}]$, where *N* denotes the N-dimensional demonstration taken, the goal of the algorithm presented in this work is to obtain a set of motion primitives or skills $S=[s_{1},s_{2},\ldots ,s_{M}]$, where *M* denotes the maximum number of segments needed to represent a specific complex task, which enables the generation of the user’s demonstration through simple movements. To achieve this, the SeGM algorithm employs a heuristic to initially estimate the critical points of change in the characteristics of the motion. For this process, the algorithm described in this work uses a custom heuristic based on two factors that can be extracted from the trajectory: abrupt changes in the second derivative and the continuity of these changes through the third derivative, or jerk. The second derivative allows us to identify abrupt changes specified by the sign change of the function. This provides an initial estimation of the changes in the direction of the functions, highlighting critical points that should be considered. By using the third derivative or jerk, the rate of change in the second derivative is analyzed, enabling the capture of changes in the system’s dynamics. This approach allows the extraction of moments where the system experiences different perturbations or control changes.

#### 3.2.1. Heuristic Segmentation Process

Prior to processing the data with the heuristic, the algorithm performs a temporal alignment process. This step is carried out when multiple demonstrations are collected for the same task, i.e., N>1. Temporal alignment serves as a preprocessing step that normalizes all demonstrations by aligning them temporally, thereby reducing potential sources of error stemming solely from the data collection process. By performing this alignment, the algorithm mitigates issues that may arise from the demonstrations being recorded at different speeds. To achieve this, the algorithm utilizes Dynamic Time Warping (DTW) [33,34].

This algorithm performs a nonlinear temporal mapping of the *N* user demonstrations against a reference value. This reference value is determined by generating the average similarity of the *N* given reference demonstrations. Within DTW algorithms, there are many ways to achieve this process. The most commonly used are methods based on Euclidean differences; however, these often yield inaccurate results by assuming a single correct way to perform a task. To avoid this potential source of error, this work adopts the Task Completion Index (TCI) metric introduced in [35]. This approach uses an index that incrementally tracks the portion of the trajectory completed up to the specific point where the task concludes. The TCI is used as a similarity measure among all initial demonstrations and is calculated for each point of each demonstration $d_{\left(i,j\right)}$, within a demonstration using the following formula:(1)$$d_{\left(i,j\right)}=\frac{\sum_{n=1}^{j}\left|x_{i,n}-x_{i,n-1}\right|}{\sum_{n=1}^{N_{i}}\left|x_{i,n}-x_{i,n-1}\right|}\;\forall j=1,\ldots ,N$$

In this equation, $x_{i,n}$ represents a point of an input demonstration at a specific time $t_{j}$, and $\left|x_{i,n}-x_{i,n-1}\right|$ denotes the distance function proposed in [35]. This function accounts for both the length of the geodesic rotation between demonstrations and the geometric distance. These factors are weighted to modulate the relative influence between them effectively.

Once the temporal alignment process has been completed, the algorithm can proceed to apply the segmentation process using the proposed heuristic. For this process, the algorithm begins by calculating the second derivative ($\frac{d^{2}}{dt^{2}}d_{i,j}\left(t\right)$), denoted as τ, and the third derivative or jerk ($\frac{d^{3}}{dt^{3}}d_{i,j}\left(t\right)$), which we denote as κ. Since the demonstrations are collected discretely as a sequence of points, both values are computed as the finite difference between sequentially centered elements. This is expressed in the following equation:(2)$$\left\{\begin{array}{c} \upsilon_{i,j}=\frac{di+1,j-di-1,j}{2\Delta t}\\ \tau_{i,j}=\frac{\upsilon_{i+1,j}-\upsilon_{i-1,j}}{2\Delta t}\\ \kappa_{i,j}=\frac{\tau_{i+1,j}-\tau_{i-1,j}}{2\Delta t} \end{array}\right.$$
where i=2,…,n−1, j=1,…,N and $\Delta t=t_{i+1}-t_{i}$. Once the jerk (κ) and second derivative (υ) values are obtained, a data normalization process is carried out. This process ensures consistency in the comparability of the values that will be used as a heuristic for obtaining the cut points. This consistency is achieved by eliminating any discrepancies that may arise from potential variations in magnitudes between different elements, allowing for more accurate detection of change points in the segments under study, thereby enhancing the robustness in detecting these points.

With the normalized values, the algorithm must make a selection of the data where the segmentation cuts need to be made. For this process, we have developed a heuristic focused on selecting those points where there is a variation in the sign υ and where the value of κ has a sufficiently large variation with respect to the surrounding points. The variation in υ (represented as δυ(t)) allows us to identify points where the curvature of the function being studied (in our case, the user demonstrations over time) has changed, thus helping to determine the areas where the local minima of the function are located. In a general form, this sign change is presented as(3)$$\delta \upsilon \left(t\right)=\mathrm{sign}\left(\frac{d^{2}d_{i,j}\left(t\right)}{dt^{2}}\right)$$

Therefore, in the algorithm, it is evaluated whether the sign of υ at time *t* changes compared to the previous time t−1. If a sign change occurs in υ, it is considered a potential segmentation point. Mathematically, this is expressed as(4)$$\mathrm{Sign}\;\mathrm{Variation}\left(t\right)=\left\{\begin{array}{cc} 1, & \mathrm{if}\;\mathrm{sign}\left(\frac{d^{2}d_{i,j}\left(t\right)}{dt^{2}}\right)\neq \mathrm{sign}\left(\frac{d^{2}d_{i,j}\left(t-1\right)}{dt^{2}}\right)\\ 0, & \mathrm{otherwise} \end{array}\right.$$

By analyzing the variation in κ (which is denoted in this work as δκ), we obtain the smoothness of these curvature changes, enabling us to discern where fluctuations in the smoothness of the data occur, allowing for a more precise analysis of these curvature change points, and even adding areas where the curvature changes are minimal but there is a change in the smoothness of the function. Since we are working with data that may vary, an algorithm has been created that operates autonomously by analyzing the jerk value in areas defined by a specific workspace. This workspace continuously traverses the function and analyzes the κ values within that window. In this way, it is not necessary to establish a fixed threshold for analyzing the values; instead, the algorithm estimates this threshold within its search window. This search window is specified by the user, allowing for varying sensitivity to small variations in the jerk generation. The algorithm uses(5)$$\delta \kappa \left(t\right)=\left|\kappa \left(t\right)-\frac{1}{2w}\sum_{k=-w}^{w}\kappa \left(t+k\right)\right|>\theta$$
where $\frac{1}{2w}\sum_{k=-w}^{w}\kappa \left(t+k\right)$ represents the average jerk at nearby points within a time interval defined by the specified search window and θ is the threshold that defines how large the variation in jerk must be to be considered significant. To avoid having the threshold manually set by the user, which could make the heuristic method less generalizable, a dynamic threshold generator has been implemented. Within the search window, the average value that the point under study must exceed is calculated. This is estimated as(6)$$\theta =\frac{1}{2w+1}\sum_{k=-w}^{w}\kappa \left(t+k\right)$$
where κ(t+k) represents the jerk value at the point t+k within the window, *w* represents the number of points to the left and right of the central point *t* in the window, and 2w+1 is the total size of the search window.

Once the values of δυ and δκ are correctly estimated, the algorithm performs a convergence between both values and selects only those segmentation points that satisfy both criteria simultaneously. This allows for an initial process of discarding points with very similar values, indicating that these points belong to the same segment. An example of the segmentation process based on the presented heuristic can be seen in Figure 3, where a 2D example of writing the letter R is shown.

#### 3.2.2. Probabilistic Merged of Segments

Although the heuristic process allows for the identification of critical cutting points, it has been observed that the generated solutions result in an excessive number of movement primitives (as seen in Figure 3c). This leads to the issue that, for a complex task, primitives are not being generated that generally encompass the learning of the specified task. This process is known as over-segmentation and prevents extracting truly representative information for the task to be performed. To address this problem, a probabilistic evaluation process was added to the algorithm to prevent this issue. This algorithm allows for grouping the segments based on their probabilistic relevance, which helps reduce heuristic relevance and establishes a series of groups that are merged based on their probabilistic weights.

To work with probabilistic models, we present an implementation based on a probabilistic clustering structure. The idea of dimensionality reduction is based on the probabilistic modeling of the critical cutting points and their combination. These cutting points are combined based on their probabilistic values. This allows points of change that are close and probabilistically similar to be grouped together in the same dataset. In this way, the combined critical cutting points are used to probabilistically estimate the remaining key points of the different demonstrations by performing a complete sampling process. This process is repeated until a series of point sets are automatically obtained. By unifying these key points, the algorithm is able to significantly reduce the number of points, thus addressing the problem of over-segmentation. A topological diagram of the probabilistic fusion process implementation can be seen in Figure 4.

For this process, the algorithm utilizes an estimation of probability densities through the use of Gaussian Mixture Models (GMMs) [36,37]. GMMs can be understood as an accumulation or weighted sum of *W* elements with different Gaussian densities. GMMs can be described by the equation(7)$$p\left(d|\lambda \right)=\sum_{i=1}^{W}w_{i}\;N\left(d|\mu_{i},\Sigma_{i}\right),$$
where *d* represents our continuous demonstration in *N*-dimensions, $w_{i}$, $i=1,\ldots ,W_{i}$, determines the values of the Gaussian mixture weights, and $N(d|\mu_{i},\Sigma_{i})$, $i=1,\ldots ,W_{i}$, define the Gaussian densities of each component. These components of the probability density are expressed as(8)$$N\left(d|\mu_{i},\Sigma_{i}\right)=\frac{1}{{\left(2\pi \right)}^{N/2}{\left|\Sigma_{i}\right|}^{1/2}}\exp\left\{-\frac{1}{2}{\left(d-\mu_{i}\right)}^{\prime }\Sigma_{i}^{-1}\left(d-\mu_{i}\right)\right\}$$
where the element $\mu_{i}$ represents the means, and $\Sigma_{i}$ defines the covariance matrix. The mixture weights must sum to 1. Therefore, in GMM models, the parameters are defined in terms of their means, covariance matrices, and the weights of the mixtures. In our algorithm, the individual weight of the critical citing points is defined as the confidence or likelihood of being at that point under the fitted GMM:(9)$$w_{i}\left(t\right)=\frac{N(d_{i}\left(t\right);\mu \left(t\right),\Sigma \left(t\right))}{\sum_{i=1}^{W}N\left(d_{i}\left(t\right);\mu \left(t\right),\Sigma \left(t\right)\right)}$$
where $d_{i}\left(t\right)$ represents the value of the point $d_{i}$ in an specific time *t*.

The use of probabilistic models has been applied in other similar approaches to estimate points of relevance in tasks and even for clustering. In all cases, a common issue arises during the initialization process and the subsequent convergence to the optimal number of clusters: determining the required number of Gaussians. In the developed algorithm, this problem is addressed through the heuristic. The critical points generated by this process are used for initialization. Once this process is completed, an iterative Expectation–Maximization (EM) process ensures reliable convergence, accurately modeling the different segments. After modeling with the Gaussians, the probability of each point belonging to specific segments must be estimated. This is achieved through a combined probability that incorporates both the expected value of the segments and the probability of belonging to the expected segment. For that purpose, we first estimate the posterior probability using Bayes theorem, where we determine the likelihood that our model estimated by the GMM (λ) has been correctly estimated.(10)$$p\left(\lambda |d\left(t\right)\right)=\frac{p(d(t)|\lambda )p(\lambda )}{p(d(t))}$$

Here, p(λ∣d(t)) represents the probability that the model generated by the GMM is appropriate for the current segment point of the demonstration d(t). The term p(d(t)∣λ) represents the reliability of the solution, specifying how likely it is to observe d(t) within our model λ. Meanwhile, p(λ) is the prior probability of the model λ, which reflects prior knowledge about how common the solution for the model is. In this case, this probability is established using the initial heuristic models. Additionally, the marginal probability p(d(t)) serves as a normalization factor for the system’s solutions. Once this process is complete, the posterior probability is maximized to obtained an optimal model $\lambda^{*}\left(t\right)$. The result of the complete segmentation process using the probabilistic approach with GMMs can be observed in detail in Figure 5. This figure presents the same example in 2D with a single demonstration, illustrating how the probabilistic fusion of the heuristic segmentation enables the segmentation process into motion primitives. Over-segmented elements are eliminated, resulting in primitives that comprehensively and meaningfully model the entire task with relevant information.

In Appendix A, Algorithm A1 is detailed, where we present the process developed for the autonomous generation of movement primitives without human supervision. The code is divided into the two main parts previously described, starting with the heuristic process and followed by the probabilistic accumulation of segments from the over-segmentation.

It is important to emphasize that our algorithm allows for a subsequent probabilistic fusion of the different types of data from the demonstrations. As mentioned at the beginning, the algorithm enables a multimodal development of the demonstrations, acquiring information not only from the position but also from other relevant data for task execution, such as joint information, task-space data, end-effector force, gripper position, etc. The process for each of these information sources follows the same steps: first, identifying a series of change points based on our heuristic, and then probabilistically modeling them. Following this, a second probabilistic fusion is performed across all the results obtained from the optimal segmentation for each case, resulting in a combined outcome from all data sources. This process ensures that, depending on the information used, the segmentation results may vary, altering task change points or segment lengths, but it does not lead to over-segmentation under any circumstances.

### 3.3. Library Generation

Once the segments have been obtained, each one is considered a new and unique primitive with no correlation to the others. Learning each segment as an individual primitive could become problematic, as it would result in an infinite number of possible primitives that essentially represent the same type of movement. This is precisely why the use of movement primitive libraries is so beneficial. These libraries enable the storage of primitives that, in essence, represent the same type of movement or skill with slight variations. Such libraries allow for restricting the learning process (for example, having a library for cooking tasks and another for cleaning tasks), providing LfD algorithms with the ability to reduce the search space when determining the optimal policy [38]. Additionally, these libraries can be expanded or reduced either by the user or the robot itself, thereby modifying the knowledge about the tasks. This adaptability enables the creation of new specific subtasks for solving larger tasks.

This is why our automatic segmentation algorithm incorporates a grouping process into different libraries of movement primitives. To perform this process, we focus on extracting two essential features for comparing the generated segments: the shape and the size of the movement. The shape allows us to establish the geometric structure of the movements, enabling an understanding of the spatial positioning of each segment. With size, we determine the amplitude required for the robot to execute the specific task. For this process, geometric comparison techniques are not suitable, as they cannot extract comprehensive information regardless of the spatial positioning of the segments. For this reason, we have developed a metric based on the probabilistic costs of transforming one segment to resemble another in both shape and size. To achieve this, we have decided to utilize Gaussian Processes (GPs) [39,40] and the Wasserstein distance [41,42] using the barycenter of the GPs, which allows us to capture both the geometric information and their variability by leveraging the uncertainty of the data. To evaluate both elements, we have implemented a process that combines both mathematical models to derive what is known as Gaussian Optimal Transport (GOT) [43]. This method generates an energy cost value that quantifies how effectively one segment can be transformed to match another exactly. Figure 6 illustrates the visual process carried out to represent the energy cost of transforming one GP into another. For this process, a temporal step is estimated, which is used to calculate the intermediate steps and, consequently, the energy cost at each of these study points.

#### 3.3.1. Gaussian Process Fundamentals

GPs allow representing a collection of random variables, where any finite subset follows a multivariate Gaussian distribution. Conceptually, a GP can be interpreted as a probabilistic distribution over a function f(x), where $f:R^{d}\to R$. For a finite set of input data $\left\{x_{1},x_{2},\ldots ,x_{n}\right\}\subset R^{d}$, this generates a vector of function evaluations $\left[f\left(x_{1}\right),f\left(x_{2}\right),\ldots ,f\left(x_{n}\right)\right]$, which follows a multivariate normal distribution. GPs are fully defined by their mean function and covariance function:(11)m(t)=E[f(t)],(12)$$k\left(t,t^{\prime }\right)=E\left[\left(f\left(t\right)-m\left(t\right)\right)\left(f\left(t^{\prime }\right)-m\left(t^{\prime }\right)\right)\right],$$
where f(t) represents the stochastic process, m(t) denotes the expected value of the process at a given point *t*, and $k(t,t^{\prime })$ encodes the covariance between two points, capturing the uncertainty and variability of the process.

When working with GPs, prior knowledge about the function f(t) is often incorporated using, in our case, the generated segments, denoted as $D={\left\{\left(t_{k},q_{k,i}\right)\right\}}_{k,i=1}^{N}$. Here, $t_{k}$ represents the time at the *k*-th point, and $q_{k,i}$ denotes the observed value in the *i*-th dimension at $t_{k}$. Given the training data and prior assumptions, the joint distribution of observed values and latent functions at test points is also Gaussian. For a training set evaluated at points *t*, the mean vector m(t) and the covariance matrix $K(t,t^{*})$ describe the GP’s behavior. By incorporating Gaussian noise with variance $\sigma_{n}^{2}$, the joint distribution over training outputs *q* and test functions $f^{*}$ is expressed as follows [44]:(13)$$\left[\begin{array}{c} q\\ f^{*} \end{array}\right]∼N\left(\left[\begin{array}{c} m(t)\\ m(t^{*}) \end{array}\right],\left[\begin{array}{cc} K\left(t,t\right)+\sigma_{n}^{2}I & K(t,t^{*})\\ K(t^{*},t) & K(t^{*},t^{*}) \end{array}\right]\right),$$

The density of $q_{i}$ for each dimension can be expressed as(14)$$p\left(q_{i}|t\right)=N\left(m_{i}\left(t\right),K_{i}\left(t,t\right)+\sigma_{n,i}^{2}I\right).$$

To optimize the GP for a set of specific segments, hyperparameter tuning is required. These hyperparameters $\theta_{i}$ influence the kernel function $k_{i}\left(\;\cdot \;,\;\cdot \;\right)$, which governs the GP’s structure. Selecting an appropriate kernel is crucial, as an incorrect choice may lead to poor task performance. In our work, we have decided to use the squared exponential kernel [45]. This kernel provides smooth and differentiable trajectories, ideal for our application. Therefore, with the use of GPs, we can model each series of segments, which may consist of one or *N* demonstrations. This enables us to calculate the energy cost required to align these GPs, making them equivalent to each other. This alignment relies on the Wasserstein distance, which provides a comprehensive metric for comparing GPs by considering both their mean and covariance structures.

For simplicity in the equations and for the reader’s convenience, the term for the mean previously represented as $m_{i}\left(t\right)$ will now be denoted as μ, and the term for the covariance previously represented as $K_{i}\left(t,t\right)+\sigma_{n,i}^{2}I$ will now be denoted as Σ. This change aims to facilitate the understanding of the equations and prevent them from becoming overly lengthy or difficult to comprehend.

#### 3.3.2. Wasserstein Distance Fundamentals

The Wasserstein distance is widely used for comparing probability distributions by taking into account their geometry (covariance) and positioning (mean). For two Gaussian Processes $N(\mu_{1},\Sigma_{1})$ and $N(\mu_{2},\Sigma_{2})$, the 2-Wasserstein distance is defined as(15)$$W_{2}\left(N\left(\mu_{1},\Sigma_{1}\right),N\left(\mu_{2},\Sigma_{2}\right)\right)={∥\mu_{1}-\mu_{2}∥}_{2}^{2}+\mathrm{tr}\left(\Sigma_{1}\right)+\mathrm{tr}\left(\Sigma_{2}\right)-2\;\cdot \;\mathrm{tr}\left({\left(\Sigma_{1}^{1/2}\Sigma_{2}\Sigma_{1}^{1/2}\right)}^{1/2}\right),$$
where $∥\mu_{1}-\mu_{2}{∥}_{2}^{2}$ measures the difference between means, and tr(·) denotes the trace of a matrix that uses the trace terms associated with $\Sigma_{1}$ and $\Sigma_{2}$ that captures the similarity in covariance structures. Using this metric, the measurement integrates both geometric and positional differences into a unified framework, making it ideal for evaluating the cost of transforming one GP into another.

To perform this comparison using the Wasserstein distance, the barycenter of the GPs to be compared was chosen. The use of this barycenter was selected for several reasons. First, it captures the mean of the GPs while also incorporating variability and uncertainty through the covariance. Second, its geometric properties enable more sophisticated comparisons. Lastly, it ensures the uniqueness of the barycenter and convergence when approximating GPs using finite-dimensional representations [46]. The barycenter is the distribution that minimizes the weighted sum of Wasserstein distances to the given GPs. We express this mathematically as(16)$$P=\underset{\mu \in P_{2}\left(H\right)}{\mathrm{argmin}}\sum_{i=1}^{N}\xi_{i}W_{2}\left(N\left(\mu_{i},\Sigma_{i}\right),N\left(\mu ,\Sigma \right)\right),$$
where ${\left\{N\left(\mu_{i},\Sigma_{i}\right)\right\}}_{i=1}^{N}$ is the set of GPs to be combined, ${\left\{\xi_{i}\right\}}_{i=1}^{N}$ are the weights assigned to each GP ($\sum_{i=1}^{N}\xi_{i}=1$), and $P_{2}\left(H\right)$ is the space of Gaussian measures in a separable Hilbert space [47]. The barycenter’s mean is a weighted average ($\mu^{★}$) and its covariance ($\Sigma^{★}$) minimizes the energy cost of transforming the covariances of the original GPs to that of the barycenter.(17)$$\mu^{★}=\sum_{i=1}^{N}\xi_{i}\mu_{i},\;\sum_{i=1}^{N}\xi_{i}{\left({\bar{\Sigma }}^{\frac{1}{2}}\Sigma_{i}{\bar{\Sigma }}^{\frac{1}{2}}\right)}^{\frac{1}{2}}=\Sigma^{★}$$

#### 3.3.3. Gaussian Optimal Transport

Once the GPs have been defined and the barycenters of these probabilistic representations have been obtained, the process of obtaining the cost maps is carried out using the implementation of Gaussian Optimal Transport (GOT) [48,49]. This method establishes a tool for comparing Gaussian distributions by using Wasserstein distances as the basis for measuring the differences in covariance distances through the use of Procrustes. This method allows a direct comparison between the structural differences of two Gaussians, without considering their geometric positioning. In this way, GOT gives us a geometric intuition about the use of covariance, allowing for a metric that is less sensitive to scale differences and capable of aligning distributions in high-dimensional settings, in addition to being a flexible method that allows modification of the covariance of the Gaussian definitions. Therefore, our algorithm combines the transport through Wasserstein distance and a direct application of GOT. First, a filtering process based on the Wasserstein distance is performed between the GPs directly. Once this process is carried out, a refinement process is applied to compare the efficiency of the Gaussian structures using the Procrustes distance. In this way, we can make an estimate that considers the geometric structure of the segments but also their information about the movement shape of each segment, allowing us to more comprehensively search for segments that are structurally more similar, regardless of their spatial positioning and shape. The full process of our method for the library generation applying all the previous mathematical descriptions is presented in Algorithm 1:
**Algorithm 1** Generation of movement primitives libraries**Input:** List of segments S=[s1,s2,…,sN], Similarity threshold St**Output:** List of libraries *L*
 1:**for** si in *S* **do** 2:    Generation of GP: GP(μi,Σi) 3:    Barycenter: Pi=argminμ∈P2(H)∑i=1NξiW2N(μi,Σi),N(μ,Σ) 4:**end for** 5:**for** si and si+1 in *S* **do** 6:    μ★,Σ★←Pi 7:    μ★,Σ★←Pi+1 8:    Wasserstein distance: Wi←W2(GP(μi★,Σi★),GP(μi+1★,Σi+1★)) 9:    Procrustes distance: Π2(Σi★,Σi+1★)=infR:R*R=I∥(Σi★1/2−Σi+1★1/2R∥F210:    Cumulative cost: C←∑(Wi+Π2(Σi★,Σi+1★))11:    **if** C−1>=St **then**12:        Generation of new library: L←L(si,si+1)13:    **end if**14:**end for**


In this way, the generated algorithm evaluates each segment with the rest, establishing the accumulated cost relationships in the transport of one segment to another. If, after evaluating everything, this value exceeds a specific threshold, the segments are considered to belong to the same family of primitives and, therefore, the generated solution establishes that they belong to the same movement library. The threshold allows the user to decide how precise they want the created libraries to be, with values very close to zero indicating segments that are practically identical, and the closer the threshold is to 0, the greater the dissimilarity between the segments. Empirically, a value of 0.7 is recommended, as it has been shown to be a stable value for all the case studies conducted. Figure 7 visually demonstrates a comparison between two segments using the proposed method.

A cost map is generated and iteratively updated until the total cost of transforming the initial GP (orange) into the final GP (blue) is obtained. The proposed method offers several advantages over other evaluation metrics, such as its ability to handle both normalized and non-normalized data, its probabilistic processing that evaluates not only the geometric solution but also the structural form of the data, and its robustness to segments with slight variations in data collection. Furthermore, it is bidirectional, meaning that comparing $GP_{1}$ to $GP_{2}$ yields the same result as comparing $GP_{2}$ to $GP_{1}$.

A visual example of a complex trajectory demonstrating how the process of generating movement libraries is performed is shown in Figure 8. In this case, a single demonstration is presented, which has been segmented into three different segments. Each segment is iteratively compared against the others. Due to its bidirectional nature, the algorithm evaluates the segments without repeating comparisons, making the process faster as it guarantees the same result for reversed comparisons. The figure illustrates this sequence and shows the mean barycenters of each segment evaluated using our translation cost evaluation algorithm.

## 4. Experiments

After explaining the operation of the developed algorithm for automatic segmentation and movement library generation, a series of tests were conducted for both 2D and 3D tasks to assess its qualitative performance. For the 2D experiments, two distinct datasets were created. The first dataset comprised uppercase letters, including all 26 letters of the alphabet, as individual elements. The second dataset consisted of word sequences, containing 20 different words ranging from 2 to 4 letters. For both datasets, segmentation and movement library generation were performed using only the geometric position of the data. Each dataset included three demonstrations for every element to capture variability in the execution of the same sequence. Additionally, the datasets were designed to be continuous, with letters written in an italic script to ensure that all segments of the uppercase letters were seamlessly connected.

To generate the datasets and perform tests in the real environment, the *Autonomous Domestic Ambidextrous Manipulator* (ADAM) robotic platform [50] was used. This platform is conformed by two UR3 manipulators as well as two Inspire Robots hands.

By leveraging the robotic platform for dataset generation, additional data such as the robotic hand’s position and end-effector force measurements could be collected. These multimodal data were integrated into the segmentation process using our SeGM algorithm, which supports advanced multimodal segmentation capabilities. For data collection via kinesthetic teaching, the robotic arms were operated in a weight-compensation mode. This mode allows users to move the robotic arm freely, preventing it from being affected by its own weight. For 2D tasks, a whiteboard was used to record demonstrations through drawing, whereas 3D tasks were conducted on a table to facilitate task-specific interactions. A visual example of the data collection for each dataset can be seen in Figure 9.

### 4.1. Algorithm Evaluation in Simulated Environments

To qualitatively evaluate the algorithm’s performance, we conducted various experiments to verify that it behaved as expected across the different 2D and 3D datasets. First, we began with the tests in 2D. Figure 10 provides a visual example from some cases on the dataset of letters. The image displays the segmentations without specifying the number of segments (top row) and the results of the generation of the movement primitive libraries (bottom row). The colors in the bottom row indicate which movement library each segment belongs to. Segments with the same color are considered as the same primitive.

It can be observed that the movements had similar and common primitives across all uppercase letters, as well as other movement primitive libraries that are not shared. From this figure, it can be observed that the letters R and B share two common primitives, represented by dark blue and brown colors. These primitives have been stored in a library of common primitives, which serves as a foundation for later use in learning algorithms based on movement primitives. The letter A shared one primitive with R and B, but the rest were distinct primitives that have been stored in separate libraries. As shown, the algorithm is capable of decomposing trajectories into critical points and logically storing the segments based on their similarity.

A more complex example is presented in Figure 11, this time for the writing of complete words. The words were reduced to a maximum of four letters due to the reach limitations of the robotic arm. In this case, the algorithm operates in the same manner, subdividing the words into their primitive movements (each letter). Learning these movement primitives enables the system to perform a reorganization process of the letters to write new words without the need to reteach the order of primitives it already knows.

As it is observed in the figure, just as with the letters, the algorithm segments each trajectory into primitive movements. In this case, it is understood that each trajectory consists of a primitive movement corresponding to the letters. It can be seen that among the different words, common letters such as “L”, “O”, or “U” are represented with the same color by our movement primitive generation algorithm. That indicates that they belong to the same class of movement. Conversely, the remaining letters, as they are not identical, are each represented as a distinct library of primitives.

To demonstrate its effectiveness in 3D environments, we performed a pick-and-place task on a table. The data were collected directly using the robot arm (presented in Figure 9), storing various task-related information. In this case, we saved the end-effector position and whether the hand was gripping or not. Using these data, two tests were conducted to evaluate the performance using different sources of information for the task. In Figure 12, the first solution shows the pick-and-place process of stacking a Rubik’s cube onto a bottle, segmented and with libraries created solely based on the end-effector position data. The second example shows the same dataset for the same task, but segmented using multimodal information from both sources that was probabilistically fused.

As observed, adding the gripper grasping information changes the segmentation. The initial segment (red) was divided into two movements where there was only one before, as the first part was performed with an open hand, and the second part with the hand holding the object. This led to changes in the movement primitive libraries, reducing the primitives from three types in the case of using only position to two primitives when using the combined information of position and grasping. In both cases, none of the solutions resulted in over-segmentation.

### 4.2. Algorithm Evaluation on a Real Robotic Platform

To assess the feasibility of using our algorithm in real-world tasks, we used the ADAM robotic platform to solve various 2D and 3D tasks after running our algorithm. To evaluate its efficiency, we conducted a series of experiments where, after segmenting and creating the different movement primitives, a new task was performed by rearranging the subtasks. This approach aims to verify the algorithm’s capabilities in segmentation and movement primitive library creation. It evaluates whether the information generated through these processes can be effectively utilized. The goal is to learn new tasks using LfD algorithms that rely on these data.

Tests were conducted in both 2D and 3D using the generated datasets. For the first 2D task, which involved writing uppercase letters, the objective was to write a series of letters by reversing the direction of the learned primitives (i.e., writing the letters backward). In the word dataset, after segmenting and learning the individual letters that composed the words, the letters were rearranged to form new words without explicitly learning their sequence. For the 3D task, a colored block stacking activity was performed. The dataset involved arranging blocks into towers of various shapes, following a consistent process for each color. After segmenting and generating the different primitives, a new task was conducted by arranging the blocks into new patterns with distinct color sequences. This approach evaluated the algorithm’s ability to generalize tasks using the learned primitives, eliminating the need to retrain the algorithm for each new task, as the movement primitives were already available. All these results for the different experiments using the ADAM robotic platform can be seen in Figure 13, where frames of the performed tasks are presented and explained in greater detail in the following video: https://youtu.be/A3m2sbUI5F0, accessed on 17 December 2024.

### 4.3. Evaluation and Comparison of SeGM’s Efficiency Against Segmentation and Clustering Datasets

In order to demonstrate the correct functioning of the segmentation process through the heuristic method and the probabilistic combination to generate movement primitives and their subsequent encapsulation in movement libraries, we conducted a quantitative comparison with various widely used state-of-the-art algorithms for the unsupervised segmentation of tasks. Our method was compared with four other algorithms for segmenting complex tasks. The first is the non-parametric segmentation algorithm, Beta Process Autoregressive Hidden Markov Model (BP-AR-HMM) [51,52]. This method has been widely used for human activity recognition and its segmentation into different tasks. The BP-AR-HMM algorithm combines Hidden Markov Models with autoregressive techniques, enabling the modeling of time series with hidden states. It is based on the use of the Beta Process, which allows it to non-parametrically estimate the optimal number of states needed without predefining them, thus enabling the segmentation of a complex trajectory into sub-trajectories. To generate the best results, we have used an autoregressive order of two. The second algorithm we compared against is the Octal Window Segmentation (OWS) [53]. This algorithm focuses on estimating and processing the error signal generated by measuring the deviation of the points in the analyzed segment with respect to an octal window. This algorithm is flexible, as it allows adaptation to different types of trajectories through the adjustment of the interpolation kernel. The third algorithm is the Sliding Window Segmentation (SWS) [54]. This unsupervised algorithm is a modification of OWS, similarly utilizing the deviation error of an octal window but, in this case, applied to imaginary interpolation. In both methods, efforts were made to optimize the algorithms as effectively as possible. To achieve this, solutions with different window sizes were generated until the best possible result for the case study was obtained. Finally, we compared our approach against Warped K-means (WKMs) [55]. This unsupervised segmentation algorithm leverages the well-known K-means method and presents a generalized approach for performing segmentations and clustering. The algorithm focuses on applying a minimization criterion for the sum of squared errors while performing a hard sequencing process that imposes a set of constraints, enabling grouping during the classification process.

To perform the comparisons, datasets generated for the study tasks were used, namely, uppercase letter writing, cursive word writing, and 3D tasks. In this case, only end-effector positional information (i.e., Cartesian data only) was utilized, as none of the other algorithms support multimodal data. Additionally, the best possible versions of each algorithm were considered to ensure fair and equitable comparisons. For this purpose, the parameter values were selected to optimize each algorithm as effectively as possible for each comparison scenario. To compare the different results, various metrics were applied to measure aspects such as the number of segments, accuracy, precision, recall, F1-score, and segment quality for the creation of primitive libraries. A ground truth was utilized for each of the datasets used. This dataset was created for both the segmentation and clustering processes. Divisions were established for each of the letters, words, and 3D tasks, specifying the expected cut points and, therefore, the number of segments to generate in each algorithm. However, the number of segments was not provided to any of the algorithms. For the library creation, segments deemed identical were grouped beforehand, and subsequently, the success rate of the proposed algorithm in this work was evaluated in terms of correctly grouping the movement primitives.

First, tests were conducted to establish classification metrics using standard measures. In our case, we used accuracy, recall, and the F1-score metric. For this purpose, we estimated the number of true/false (T/N) positives/negatives (P/N) for the segments generated by our algorithm. Based on this, we define accuracy as the number of correctly predicted segments (denoted as $S_{c}$) compared to all the segments specified in our ground truth ($S_{gt}$). The precision of the solutions for each algorithm is understood as the number of true positives relative to the total number of positive predictions, while recall is defined as the number of true positives relative to the sum of true positives and false negatives. Finally, we use the F1-score, which is expressed as twice the product of precision and recall divided by the sum of both. Each of the equations used for the different metrics can be seen in the following equations:(18)$$\begin{array}{cccc} \mathrm{Accuracy} & =\frac{S_{c}}{S_{gt}}, & \;\;\;\mathrm{Precision} & =\frac{TP}{\left(TP+FP\right)}, \end{array}$$(19)$$\begin{array}{cccc} \mathrm{Recall} & =\frac{TP}{\left(TP+FN\right)}, & \;\;\;F1 & =2\times \frac{\left(precision\times recall\right)}{\left(precision+recall\right)} \end{array}$$

We have compared each of the unsupervised segmentation algorithms against all the datasets. The results are detailed in Table 1.

As observed in the results from the table, the SeGM algorithm developed in our work is able to achieve much better results than the other compared algorithms. This is because our algorithm segments the least and produces segments that are almost identical to the ground truth. As seen, our algorithm achieves at least a 95% accuracy, a minimum precision of 97.1%, a minimum recall of 98.1%, and an F1-score value of 97.7%. In all the studied datasets, our algorithm generates better solutions than the other algorithms, with WKM being the second-best performer, but still far behind the capabilities of the SeGM algorithm presented in this work.

After analyzing the standard classification metrics, we conducted a comparison to evaluate the quality of the learned segments. For this, we performed a Leave-One-Out Cross-Validation (LOOCV) process for each of the generated segments. Each segment was treated as a validation set, while the remaining generated segments were considered as training sets. Using these training sets, we constructed a dictionary of segments to evaluate the performance for each subset. If the movement subgroup being validated differed from what was recorded in the reference dictionary of all segments, it was counted as an error. To further assess the classification of each segment, the expected segment was also recorded. The results of applying LOOCV can be seen in Table 2.

As can be observed, the LOOCV calculated for our SeGM method shows an improvement compared to the other algorithms, demonstrating that the quality of learning the segments (or motion primitives) is more optimal than those generated by the rest of the algorithms. Furthermore, it is evident that our method is the only one that approaches (and slightly surpasses) the results of the expected segmentation for each of the study datasets. Additionally, it can be noted that the solutions provided by the SeGM are much more stable than those of the other algorithms. This is evident in the fact that the results for the 2D and 3D datasets experienced only slight variations, whereas the other algorithms exhibited a significant performance drop in 3D tasks compared to 2D tasks. A visualization of the performance and segmentation solutions provided by each of the compared algorithms against some examples from the generated datasets can be seen in Figure 14, where each color represents the different segments obtained by each method. It is evident that the SeGM algorithm presented in this work produces the least amount of over-segmentation and achieves results that decompose complex trajectories into movements that are sufficiently simple to be easily learned while still encapsulating different types of movements.

Finally, we evaluated the efficiency of the classes generated by our method after the segmentation process. For this, we compared the groupings produced by our energy-cost-based algorithm for converting one GP into another against the ground truth groupings of each dataset (the ground truth was generated by asking different users how they would group each segments). To assess this, we used accuracy, precision, recall, and F1-score by comparing the classes created for each dataset with those derived from the ground truth. These results are presented in Table 3.

As observed, our algorithm is capable of generating motion primitive libraries that closely match the ground truth, achieving values that were never below 93%. However, it is important to note that the lowest performance was observed in the case of words. This was due to the difficulty of estimating the division points between letters when dealing with cursive writing, as the endpoints of one letter and the start of the next are not always clear. Despite this, our algorithm still produces motion primitive libraries that are highly similar to those generated as ground truth.

## 5. Conclusions and Future Works

The use of motion primitives for Learning from Demonstration is one of the most widely adopted techniques today. However, the complexity of capturing information for simple tasks in a way that allows for subsequent combination limits the efficiency of certain processes. To address this issue and enable its application to LfD techniques, we propose an algorithm for acquiring complex tasks and generating motion libraries. This algorithm consists of an unsupervised segmentation approach based on two steps: an initial heuristic method for generating cut points and a second step that performs probabilistic fusion of the initial segments using Gaussian Mixture Models. By combining these methods, the algorithm avoids the over-segmentation often observed in approaches relying solely on heuristic processes. The second objective of the algorithm is the creation and comparison of motion libraries. For this purpose, Gaussian Processes are utilized to provide a probabilistic encapsulation of the individual segments or groups of segments. These GPs are compared against one another by evaluating the energy cost of transforming one GP into another. This evaluation employs the Wasserstein distance and Gaussian Optimal Transport, enabling the assessment of similarity between segments not only through geometric processes but also by considering the movement’s structure.

With this approach, the SeGM algorithm proposed in this work is capable of encapsulating motion data from an entire demonstration, creating a motion model that defines distinct primitives usable in any LfD algorithm. The main advantages of the proposed algorithm include its ability to work in N dimensions regardless of the number of demonstrations, its capability to probabilistically combine diverse sources of information (such as end-effector position, gripper force, etc.), and its independence from user input, such as specifying the number of segments to find, making it entirely automatic.

To evaluate its efficiency, comparisons were made with four unsupervised segmentation algorithms using three custom datasets for 2D and 3D tasks. The SeGM algorithm developed in this work showed improvements in all efficiency metrics compared to other algorithms, producing better segmentations and avoiding over-segmentation in all study cases. Its effectiveness was also tested against clustering datasets for the different segments to create motion primitives, achieving a high success rate, always above 93% compared to the generated ground truth. Additionally, its efficiency was tested in real tasks, such as 2D handwriting, where the goal was to use the learned primitive libraries to perform tasks by reordering the segments or writing words not explicitly demonstrated. In 3D, a cube stacking task was performed, modifying the stacking order using the primitive libraries learned from a single continuous demonstration. In both cases, the robot successfully utilized the previously generated motion libraries, producing satisfactory results.

For future work, we propose extending the algorithm to include task understanding by adding semantic labels to each motion primitive library and expanding its application to bimanipulation tasks. This would allow data to be directly extracted from the robot’s perception, eliminating the need to use the robot’s arm for demonstrations.

## Figures and Tables

**Figure 1 biomimetics-10-00064-f001:**
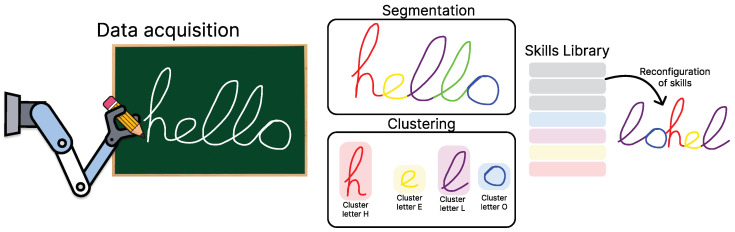
General scheme of the **Segmentation and Grouping Model (SeGM)** algorithm presented in this work. Our algorithm consists of two main components. The first is the segmentation of the collected demonstrations, which allows for identifying critical cutting points (points used to segment the entire task into subtasks) using a heuristic that takes advantage of changes in the smoothness of the movements. After this, a clustering process based on probabilistic methods is performed, where the energy cost (based on how likely the segments are to be similar) to transform one segment into another is computed. This approach enables the creation of a library of movements (or skills) that can be reconfigured to generate new movements without the need to learn them from scratch.

**Figure 2 biomimetics-10-00064-f002:**
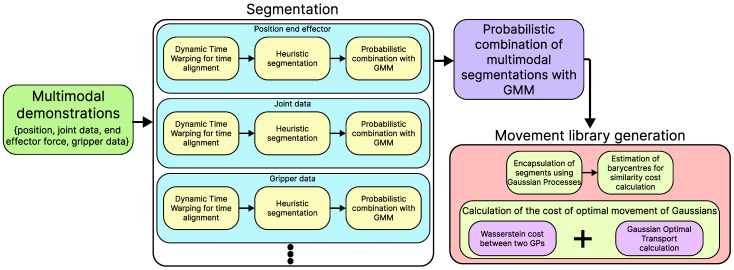
General algorithm of the processes and subprocesses performed by our approach. The algorithm begins with demonstrations that may be multimodal (i.e., derived from various sources such as the end-effector position, grip force, hand position, etc.). For each case, a segmentation process is carried out, including temporal alignment, an initial heuristic estimation of the segments, and a probabilistic combination using GMMs. If different segmentation sources are present, they are probabilistically combined using the same GMM process. After segmentation, the generation of the movement libraries begins. Segments are encapsulated using Gaussian Processes (GPs), over which their barycenters are estimated to calculate the total cost of transforming one segment to resemble another. This is achieved through the computation of the Wasserstein distance and Gaussian Optimal Transport. Iteratively, an energy cost is obtained, which determines how similar and efficient two segments are. Based on this value, the various libraries of primitives are created.

**Figure 3 biomimetics-10-00064-f003:**
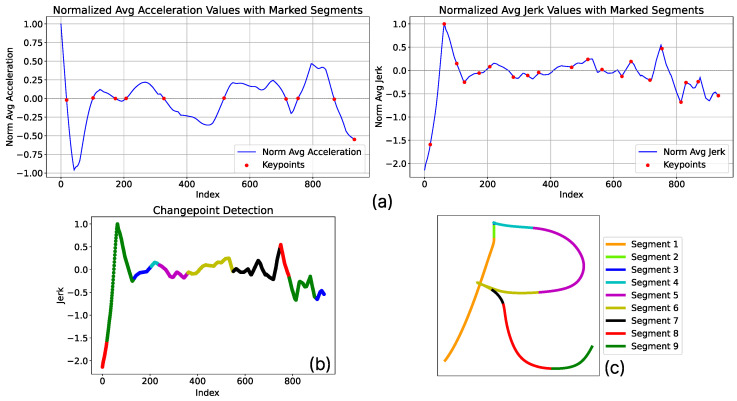
Example of results obtained through the application of the heuristic for segmentation. (**a**) Relevant points obtained after applying the sign variation values of δυ and the values of δκ using a window size of 10 units. (**b**) Result after combining both criteria on the jerk graph. It is observed that the number of values obtained solely from jerk is reduced by using sign changes in the second derivatives. (**c**) Final segmentation obtained using only the results of the heuristic. It is observed that solely using this heuristic leads to over-segmentation, creating primitives that do not contain sufficiently relevant information to be considered representative.

**Figure 4 biomimetics-10-00064-f004:**
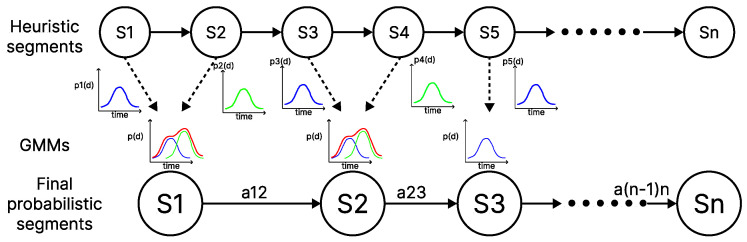
Topological diagram of the segmentation process using GMMs. The heuristic segments serve as initialization points for the probabilistic blending, thereby reducing unnecessary segments.

**Figure 5 biomimetics-10-00064-f005:**
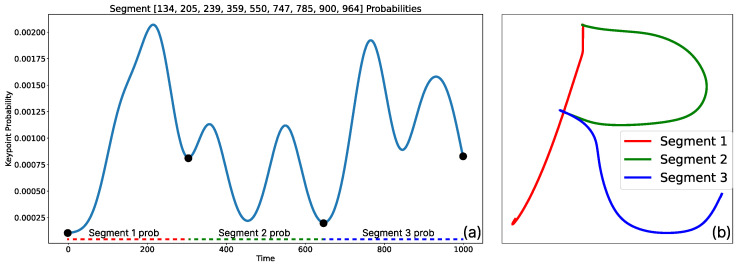
Complete solution after the probabilistic fusion process for the over-segmented data. (**a**) Representation of the normalized probability obtained after the GMM generation process and its combination. The black points represent the final keypoints on the probabilistic data, marking the locations where changes occur in the estimated common probabilities of the points. (**b**) Representation of the example “R” in 2D. After the process, the algorithm successfully generates three segments that encompass the representation of the entire task.

**Figure 6 biomimetics-10-00064-f006:**
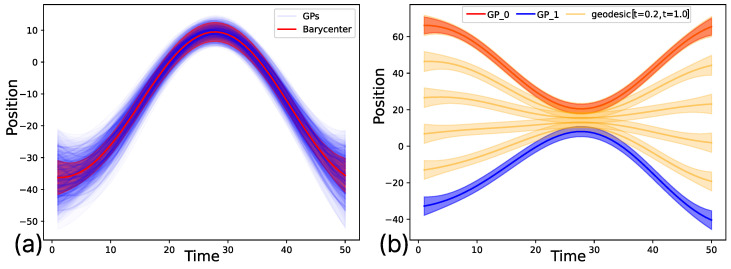
Application of the cost calculation algorithm for the generation of movement primitive libraries. (**a**) Example of barycenter estimation (red) compared to the GPs of the demonstrations of the same segment (blue). The barycenter captures the mean value (red line) as well as the weighted cumulative sum of the covariances (red shading). (**b**) Visual example of the accumulated cost of transforming one GP barycenter (red) into another (blue). Each yellow GP represents a movement with a spatiotemporal interval of t=0.2 s, used to calculate the cumulative value of each movement. The result of this calculation, obtained through the Wasserstein distance and the GOT method, allows us to determine whether two segments are similar or not.

**Figure 7 biomimetics-10-00064-f007:**
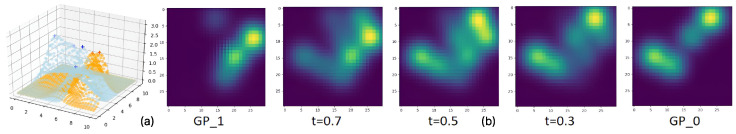
Data processing for evaluating the similarity between two segments. (**a**) Spatial representation of the segments to be compared. (**b**) Processing steps to compute the total cost of transforming one segment into the other. It can be observed how the mounds (highlighted in green on the height map) shift progressively to match the shape of the other segment under evaluation. The accumulated transport cost is used to estimate the similarity: the more energy required, the less similar the two segments are.

**Figure 8 biomimetics-10-00064-f008:**
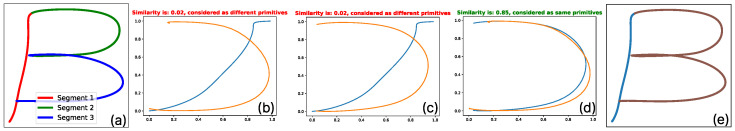
Example of evaluating the movement primitive libraries. (**a**) Initial segmentation, (**b**) normalized comparison of segment 1 against segment 2, which do not belong to the same primitive library, (**c**) normalized comparison of segment 1 against segment 3, which do not belong to the same primitive library, (**d**) normalized comparison of segment 2 against segment 3, which belong to the same primitive library, and (**e**) representation of each segment according to the library they belong to, where the same color indicates they are part of the same primitive library.

**Figure 9 biomimetics-10-00064-f009:**
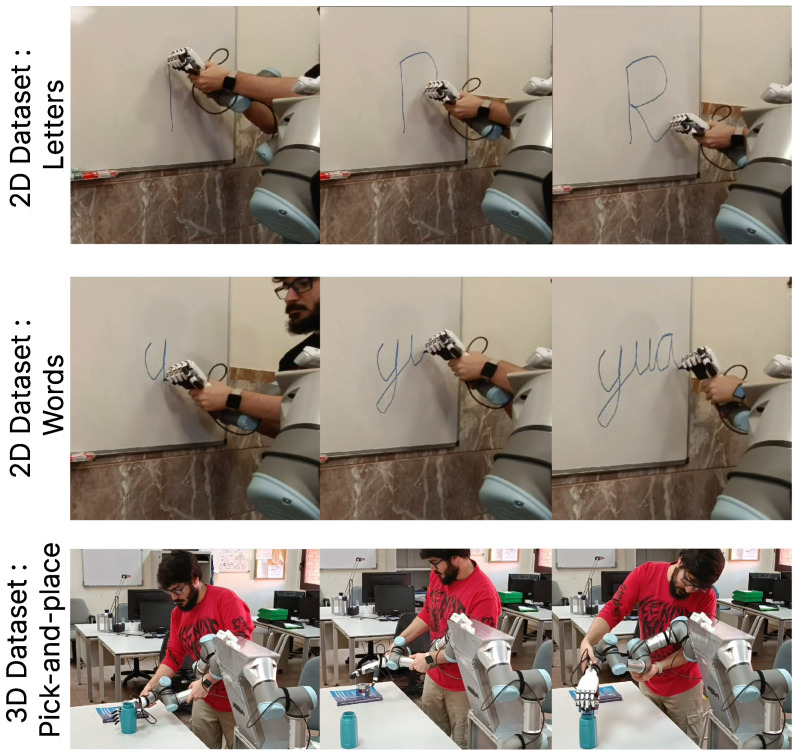
Dataset generation with the arms of the ADAM robot. The first and second rows show examples of the 2D dataset cases, while the third row presents an example of the 3D dataset for a pick-and-place task.

**Figure 10 biomimetics-10-00064-f010:**
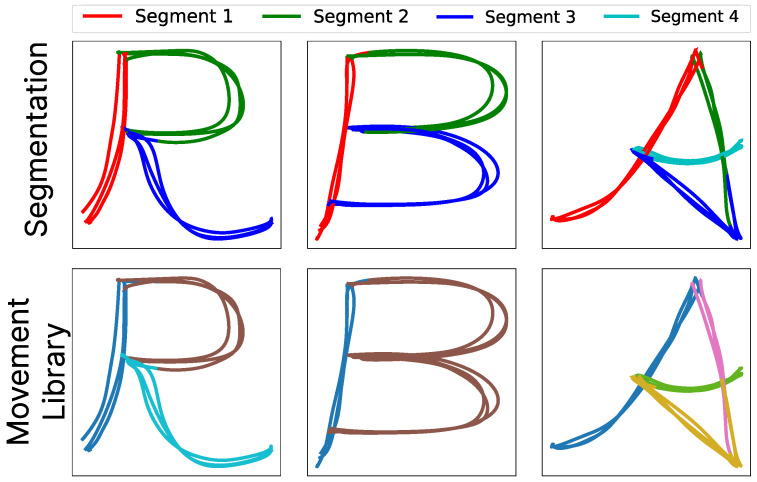
The top row represents the final segments obtained after the probabilistic segmentation process for each letter using GMMs, while the bottom row represents the process of creating movement primitives. Common colors indicate movements that belong to the same primitive library.

**Figure 11 biomimetics-10-00064-f011:**
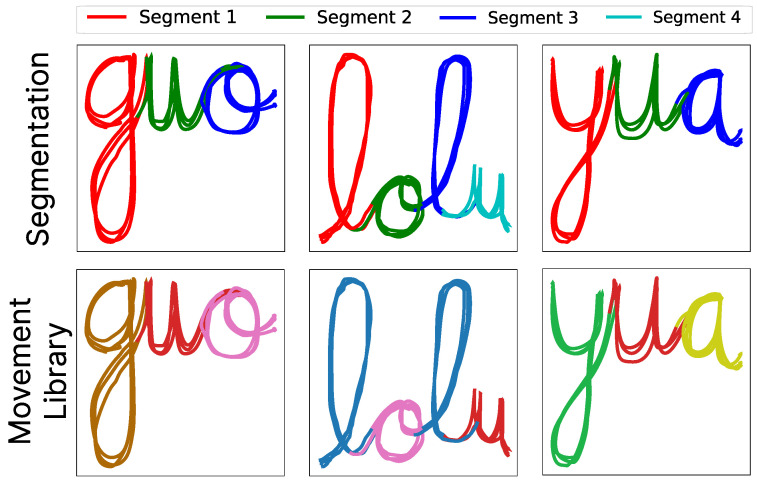
The top row represents the final segments obtained after the probabilistic segmentation process for each word using GMMs, while the bottom row represents the process of creating movement primitives. Common colors indicate movements that belong to the same primitive library.

**Figure 12 biomimetics-10-00064-f012:**
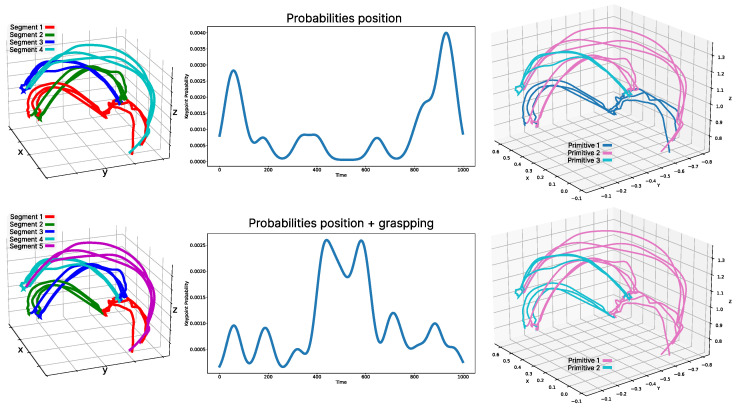
Example with single and multiple sources of information. The top row shows an example of a pick-and-place task segmented and clustered using only the end-effector position. The bottom row shows the same task but using the gripper information to indicate whether it is holding an object or not. Both the segmentation and clustering differ in each case.

**Figure 13 biomimetics-10-00064-f013:**
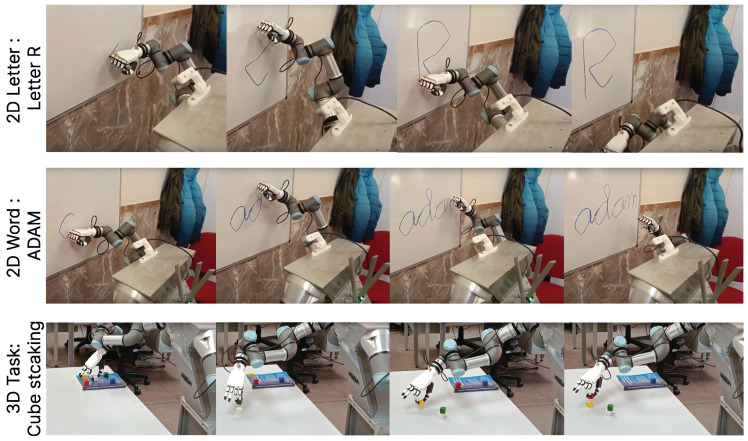
Example of tasks performed after completing the segmentation process and creating movement primitive libraries using the SeGM algorithm. With the knowledge obtained, we are able to reorganize the subtasks to create new complex skills without the need to teach them again.

**Figure 14 biomimetics-10-00064-f014:**
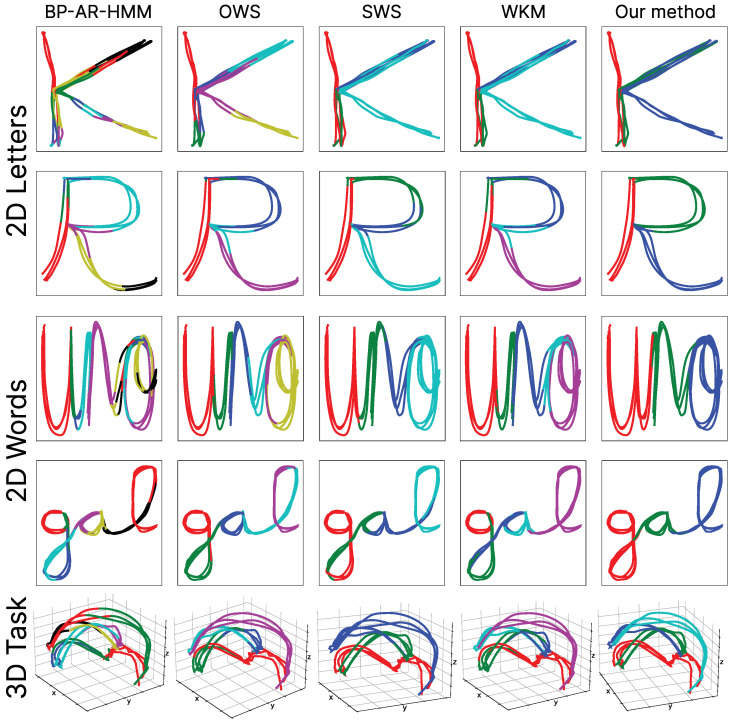
Example of segmentation results by the different compared algorithms for 2D letters (ground truth for both cases, 3 segments), 2D words (ground truth for both cases, 3 segments), and a 3D pick-and-place task (ground truth, 4 segments).

**Table 1 biomimetics-10-00064-t001:** Segmentation metric comparison results.

Dataset	Metric	BP-AR-HMM	OWS	SWS	WKM	Our Method
*2D Letters*	Accuracy (%)	75.3	79.2	85.4	89.4	**99.8**
	Precision (%)	65.4	71.8	82.6	81.2	**99.3**
	Recall (%)	71.2	74.6	88.9	87.4	**100**
	F1 (%)	68.2	73.1	85.6	84.2	**99.6**
*2D Words*	Accuracy (%)	69.4	73.1	77.4	77.2	**95.7**
	Precision (%)	59.9	68.2	70.2	76.5	**97.1**
	Recall (%)	62.3	69.5	72.1	77.9	**98.4**
	F1 (%)	61.1	68.8	71.1	77.2	**97.7**
*3D Tasks*	Accuracy (%)	64.2	70.5	74.3	76.6	**95.5**
	Precision (%)	50.3	67.4	71.5	75.3	**97.4**
	Recall (%)	53.9	68.3	73.1	78.1	**98.1**
	F1 (%)	52.1	67.9	72.3	76.7	**97.7**

**Table 2 biomimetics-10-00064-t002:** LOOCV segmentation performance. Expected by ground truth vs. expected by algorithms.

Dataset	Metric	BP-AR-HMM	OWS	SWS	WKM	Our Method
*2D Letters*	Accuracy (%) (Expected Alg.)	68.42	71.54	88.49	87.24	**97.45**
	Accuracy (%) (Expected Segm.)	96.87				
*2D Words*	Accuracy (%) (Expected Alg.)	68.12	70.48	88.79	89.36	**96.88**
	Accuracy (%) (Expected Segm.)	96.45				
*3D Tasks*	Accuracy (%) (Expected Alg.)	59.68	70.32	77.45	78.69	**93.96**
	Accuracy (%) (Expected Segm.)	93.74				

**Table 3 biomimetics-10-00064-t003:** Metrics resulting from the SeGM clustering.

Dataset	Accuracy (%)	Precision (%)	Recall (%)	F1 (%)
*2D Letters*	98.7	96.4	95.8	96.1
*2D Words*	93.6	94.1	93.8	93.9
*3D Tasks*	98.4	96.2	95.1	95.6

## Data Availability

The code of the proposed method is fully accessible from https://github.com/AdrianPrados/Segmentation-and-Grouping-Model, accessed on 18 December 2024.

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
