# Peer review of "Segment, Compare, and Learn: Creating Movement Libraries of Complex Task for Learning from Demonstration"

_biomimetics, 2025, doi:10.3390/biomimetics10010064_

Round 1
Reviewer 1 Report
Comments and Suggestions for Authors
Summary:
The paper considers the problem of learning robotic skills using learning from demonstration. The authors are arguing that one possible approach to achieve this is to decompose the tasks into shorter movement primitives. Then, if these are learned, they can be assembled into a larger number of tasks.
To facilitate this process, the authors propose a technique to automatically segment demonstrations into parts that would correspond to the movement primitives. These parts are then clustered into groups based on the energy necessary to transform one movement part into another. This distance can also be modeled as a probabilistic cost.
Strengths:
* Clear description of the objectives of work.
* The work is implemented on a physical robot, source code is available.
* The proposed approach builds on intellectually sophisticated technologies.
* The proposed work outperforms the chosen baselines along the performance criteria chosen by the authors.
Weaknesses:
* There are a number of orthographical and grammar errors in the paper.
* The detailed introduction of Gaussian Processes, Wasserstein distance and optimal transport is not necessary. This is background information that the reader, if needed, can easily look up elsewhere.
Author Response
Comments 1: There are a number of orthographical and grammar errors in the paper.
Response 1: Thank you very much for your comments regarding the grammar and spelling in our work. We agree with your observations about the presence of certain grammatical errors throughout the explanations of our work. To address this, we have conducted a thorough review of the English. We have made changes to expressions, corrected typographical errors, and added phrases that are more appropriate for clear understanding. These corrections have been applied throughout the entire paper.
Comment 2: The detailed introduction of Gaussian Processes, Wasserstein distance and optimal transport is not necessary. This is background information that the reader, if needed, can easily look up elsewhere.
Response 2: Thank you very much for your comment regarding the mathematical background of our method. We completely agree with your point of view and have noticed that the mathematical introductions can be too extensive for the work presented, and they could be revisited in more detail in previously published papers.
Nonetheless, we believe that having a brief overview of the mathematical concepts within the same paper in which they are applied can be of slight help to the reader, to avoid the need to consult a wide variety of literature in this field whenever there is a small doubt about the presented processes. Therefore, in light of your comment, we have decided to substantially reduce these mathematical implementations or even eliminate them completely. To do this, we have condensed the mathematical explanations of the two main elements of our work, which are the Gaussian processes and the Gaussian Mixture Models. We have reduced the equations presented and rewritten the paragraphs to remove information that can be found in other papers. As for the paragraphs on Wasserstein distance, we have rewritten them but kept the equations. This is because these equations were modified for our algorithm and are necessary to estimate the correct functioning of our method (e.g., using barycenters, which is not typically used in the estimation of Wasserstein distance). For the Gaussian Optimal Transport section, the equation has been removed, and two references have been added so that the reader can look up the information regarding the removed equation. These equations have been retained in the pseudocode, which we believe is sufficient for readers to understand in detail the mathematical techniques used.
Reviewer 2 Report
Comments and Suggestions for Authors
This manuscript proposes a novel Segmentation and Grouping Model (SeGM) algorithm designed for Learning from Demonstration (LfD). The algorithm employs heuristic-based segmentation followed by probabilistic clustering using Gaussian Mixture Models (GMMs) and Gaussian Processes (GPs) to create reusable movement libraries for robotic tasks. The authors validate their approach with both 2D and 3D robotic tasks, showing promising results in creating efficient and generalizable movement primitives. The topic is relevant and timely for advancing LfD applications, particularly in environments that require adaptive and reusable robotic skills. However, while the work introduces an interesting concept, the manuscript has several issues related to methodological details, clarity of results, and overgeneralization of findings.
1.The manuscript emphasizes heuristic segmentation but provides insufficient mathematical detail on how critical segmentation points are derived. While third derivatives and Gaussian Mixture Models are mentioned, the methodology lacks rigor in explaining how these tools are combined to prevent over-segmentation.
2.The manuscript claims that the algorithm is generalizable to a wide range of tasks, but the validation is limited to specific cases (e.g., writing letters and pick-and-place tasks). The experiments lack diversity in task complexity and environmental conditions.
3.Certain terms, such as "energy cost" and "critical cutting points," are used without proper definition, which may confuse readers. Additionally, the writing contains grammatical errors and overly long sentences, which detract from readability.
Author Response
Comments 1: The manuscript emphasizes heuristic segmentation but provides insufficient mathematical detail on how critical segmentation points are derived. While third derivatives and Gaussian Mixture Models are mentioned, the methodology lacks rigor in explaining how these tools are combined to prevent over-segmentation.
Response 1: Thank you very much for your comments; they are incredibly helpful in improving our work. Regarding the explanation of the critical cutting points in the heuristic process, these are detailed both in the text following the explanation of DTW and in the pseudocode of Section A. The algorithm utilizes changes in curvature through jerk to identify these critical points. Equations 2, 3, 4, 5, and 6 describe how this process is applied mathematically, and Step 1: Heuristic keypoints generation in the pseudocode of Section A presents the implementation of the algorithm in a mathematical format, explaining in detail how it has been implemented. With these elements, we consider that the heuristic aspect of our algorithm is well defined.
Regarding the methodology of the probabilistic combination,we identified that the developed process was not adequately explained. The algorithm unifies the different cutting points obtained heuristically through a probabilistic process. Our algorithm uses the various cutting points as an initial approximation of where to position the GMMs to perform the probabilistic process. Once initialized, the algorithm evaluates how similar the probabilities defined for nearby points are. If they are relatively similar, these points are grouped within the same segment.
This process is repeated for the critical cutting points, adding all intermediate points between the different critical points. This approach drastically reduces the number of cutting points, resulting in significantly less over-segmentation. The process is performed automatically, and the algorithm stops when it reaches the number deemed necessary for the process. This process relies on the use of GMMs and the combination process based on Bayes' theorem to estimate the probabilities of the entire model.
The results are shown in detail in Figure 5, where the example previously presented in Figure 4 is revisited, but with our probabilistic segment fusion process applied. Additionally, the mathematical process has been written in pseudocode format in Appendix A. To clarify this process in the text, a paragraph has been added at the beginning of Section 3.2.2, Probabilistic Merge of Segments The added text is as follows:
“The idea of dimensionality reduction is based on the probabilistic modeling of the critical cutting points and their combination. These cutting points are combined based on their probabilistic values. This allows points of change that are close and probabilistically similar to be grouped together in the same data set. In this way, the combined critical cutting points are used to probabilistically estimate the remaining key points of the different demonstrations by performing a complete sampling process. This process is repeated until a series of point sets are automatically obtained. By unifying these key points, the algorithm is able to significantly reduce the number of points, thus addressing the problem of over-segmentation.”
Comment 2: The manuscript claims that the algorithm is generalizable to a wide range of tasks, but the validation is limited to specific cases (e.g., writing letters and pick-and-place tasks). The experiments lack diversity in task complexity and environmental conditions.
Response 2: Thank you very much for your comment regarding the tasks that our algorithm can address. First, we would like to acknowledge your perspective, as it has been mentioned several times in the text that our algorithm can be used for various tasks. The fact that we have used only three tests (two of them in 2D) and the third in a pick-and-place task is related to the comparisons between the different algorithms presented in our work. Since we decided to perform comparisons in both 2D and 3D, not all the algorithms used could handle highly complex tasks. To ensure a fair comparison and obtain results that accurately reflect the study, we decided to limit these comparisons to widely used tasks in the field of learning from demonstration, such as handwriting tasks or pick-and-place tasks.
That said, the tasks performed in both cases exhibit considerable complexity. For the 2D tasks, writing cursive words is complex both in task execution and in the process of selecting the correct cutting points. Regarding the 3D task, the pick-and-place process involves stacking, and as shown in the video, this process is highly complex due to the precision required for cube alignment and the arm positioning for correctly grasping each cube. Considering these factors, we believe the experiments conducted are valid, as they involve tasks that serve as benchmarks for testing various algorithms employing similar techniques.
Additionally, in Section 4.3, the algorithm's functionality was also tested using only position data, as well as probabilistically combining position data with information about whether an object was being held. This demonstrated how such information could influence the segmentation process if necessary.
However, we recognize that there is a wide range of additional tests that could be conducted, particularly for more complex tasks (e.g., cooking tasks). Since the primary goal of this work is to demonstrate the efficiency and functionality of the developed algorithm, we consider the presented experiments sufficient to highlight its advantages compared to other algorithms. Nonetheless, as we intend to continue this line of research, future work will involve applying this technique to more complex tasks, integrating it with learning-from-demonstration techniques to showcase its continuous functionality alongside a learning process.
Taking all this into account, the tasks presented serve as an initial proof to demonstrate the functionality and, above all, the efficiency of the algorithm compared to other methods. In the future, its performance could be evaluated in more complex tasks, both in terms of task execution and temporal complexity.
Comment 3: Certain terms, such as "energy cost" and "critical cutting points," are used without proper definition, which may confuse readers. Additionally, the writing contains grammatical errors and overly long sentences, which detract from readability.
Response 3: Thank you very much for your comments regarding the grammatical errors and undefined terms. We have noticed that, as pointed out in your comment, there were irregularities in the explanation of the terms "Energy cost" and "critical cutting points." To address this issue, we have added a brief explanation in the text the first time these terms are presented (which occurs in the description of Figure 1). The corrected text in that image is as follows:
“General scheme of the Segmentation and Grouping Model (SeGM) algorithm presented
in this work. Our algorithm consists of two main components. The first is the segmentation of the collected demonstrations, which allows for identifying critical cutting points (points used to segment the entire task into subtasks) using a heuristic that takes advantage of changes in the smoothness of the movements. After this, a clustering process based on probabilistic methods is performed, where the energy cost (based on how likely the segments are to be similar) to transform one segment into another is computed. This approach enables the creation of a library of movements (or skills) that can be reconfigured to generate new movements without the need to learn them from scratch.”
Additionally, these terms have been explained and highlighted in italics in the part where the development of our algorithm is explained in detail, so that they are described in a much clearer way for the reader. The term "critical cutting points" has been explained in detail and highlighted in Section 3.1 General description:
“To obtain the segments into which our demonstrations are divided, it is necessary to determine, what we have called, critical cutting points P. These points specifically indicate where each segment begins and ends within the complete demonstration. To achieve this, a specific heuristic has been developed, detailed in Sect.3.2. This heuristic identifies relevant points within the demonstrations based on the structure of the demonstration data.”
This process has also been carried out with the term "energy cost" in the same section:
“This enables the grouping of similar trajectories into movement primitives, thereby reducing the redundancy of libraries that may be considered identical. For this process, the algorithm employs what we have called as energy cost estimation required to transform one segment into another, considering both their shape and size. Gaussian Processes (GPs) are utilized to probabilistically estimate the variability of a single data point or a set of segments from different demonstrations. This approach establishes a cost map necessary to adjust the structure of each GP across different groups of segments.”
Regarding the grammatical errors, thank you very much for pointing them out. Since English is not our native language, we may have made some mistakes when writing. Taking your comments into account, we have conducted a thorough review of the entire submitted work, carefully reading and correcting the grammatical and syntax errors found in the text. To achieve this, we meticulously reviewed each paragraph and figure caption, identifying all possible errors and incorporating certain expressions to improve the clarity and understanding of our work. We also identified sentences that were too long. We have addressed this issue by dividing them into shorter, more concise sentences. This improves both the clarity and readability of the text.